# Effect of Inclusions on the Corrosion Properties of the Nickel-Based Alloys 718 and EP718

**Ekaterina Alekseeva [1],\*, Andrey Karasev [2],\*, Pär G. Jönsson [2]**  **and Aleksey Alkhimenko [1]**

[1]  Scientific and Technological Complex "New technologies and materials", Institute of Advanced Engineering Technologies, Peter the Great Saint-Petersburg Polytechnic University, Polytechnicheskaya 29, 194064 Saint-Petersburg, Russia; a.alkhimenko@spbstu.ru

[2]  Department of Materials Science and Engineering, School of Industrial Engineering and Management, KTH Royal Institute of Technology, Brinellvägen 23, 100 44 Stockholm, Sweden; parj@kth.se

\*  Correspondence: alekseeva_el@spbstu.ru (E.A.); karasev@kth.se (A.K.); Tel.: +7-812-552-6441 (E.A.); +46-08-790-8357 (A.K.)

**Abstract:** Inclusions in steels and alloys are known to lower the resistance to deformation, as well as to lower the mechanical, corrosion and other properties. Studies of inclusions in nickel-based alloys are important since these materials could suffer from corrosion degradation in harsh operational conditions. This, in fact, could lead to a pitting initiation around the inclusions. Two industrial Ni-based alloys (alloy 718 and EP718) were investigated to determine the harmful effects of different inclusions on the corrosion resistance of Ni-based alloys. Specifically, the inclusion characteristics (such as composition, morphology, size, number and location) were determined for inclusions collected on film filters after electrolytic extraction and dissolution of a metal matrix around different inclusions on surfaces of metal samples after electrolytic extraction (EE). It was found that both Ni-based alloys contain various inclusion types: carbides (large size NbTi-C and small multicomponent carbides), nitrides TiNb-N and sulphides (TiNb-S in EP718 alloy). The most harmful effects on the corrosion resistance of metal were detected around sulphides and small carbides containing Mo, W, Cr. Dissolution effects were also observed around large carbides and nitrides, especially around inclusions larger than 10 μm. Moreover, the dissolution of a matrix around inclusions and clusters located on the grain boundaries were found to be 2.1–2.7 times larger compared to inclusions found inside of grains of the given alloy samples.

**Keywords:** nickel-based alloys; corrosion; inclusions; oil and gas industry; electrolytic extraction; alloy 718

## 1. Introduction

Many investigations show that inclusions (incoherent precipitates), which are formed during production [1–3], can significantly affect mechanical, corrosion, deformation, machining properties of steels and alloys. This has, for example, been reported with respect to low alloyed steels and carbon steels [4–6], stainless steels [7–10], and other alloys [11–13].

Voids and stressed zones between the inclusion and a matrix represents sites where microcracks and pittings are initiated. Also, around inclusion zones are formed zone which contain no or lower concentrations of precipitated phases and elements contained in inclusions or metal zones depleted in alloying elements. As a result, the metal zones adjacent to inclusions are characterized by higher levels of defects and stresses [1]. Therefore, the mechanical properties and corrosion resistance of steels and alloys depend on the amount of these harmful inclusions, as well as on their size, morphology and distribution.

Ni-based alloys have a high corrosion resistance and good mechanical properties. Thus, they are widely used in industrial applications as a material for wellhead components and downhole equipment for corrosive service at high temperatures and pressures in the presence of aggressive components (such as $H_2S$, $CO_2$, chlorides) [14]. Since Ni-based alloys have complicated chemical compositions, it leads to a formation of various deleterious inclusions and phases such as carbides, sulphides, delta-phase, etc. [15]. Some inclusions can initiate a pitting corrosion and could lead to environmentally assisted cracking, corrosion fatigue and hydrogen embrittlement. Therefore, studies of corrosion and mechanical properties of Ni-based alloys and their dependence on different inclusion types has been the object of attention of many previous studies [16–20].

For instance, it was reported in [21,22] that stable pittings started at Nb-C rich particles, whereas nitrides of Ti and Nb have a lower effect on corrosion [23]. Precipitation of carbides containing chromium and tungsten (such as $M_{23}C_6$, $M_6C$) on the grain boundaries is also possible precursor sites for pitting in Ni-base alloys [24,25]. Mo-rich phases are also known to lead to sensitization and poor corrosion properties on the grain boundaries [26]. In [27] it was shown, that a delta phase has a detrimental effect on the passivation and potential breakdown of alloys. Moreover, the delta phase is also a trapping site for hydrogen, which lowers the resistance to hydrogen embrittlement and which is dependent on the morphology of second phase particles [28,29]. Despite high crystallization rate additive technologies result in specific inclusions, such as coarse carbides of Ti, Nb, Cr, Mo, Nb/Mo-rich particles [30,31].

Overall, it is clear that the Ni-based alloys contain various inclusion types. However, at the moment there are unclear and contradictory data on the influence of composition, size, location and morphology of different inclusions on the corrosion properties. Therefore, the main goals of the present study are to test an application of the electrolytic extraction technique for precise three-dimensional investigations of different inclusions and clusters (their composition, morphology, size, number and location in the matrix) in present two industrial Ni-based 718 and EP718 alloys. Furthermore, to evaluate the effect of various inclusions and clusters on the corrosion resistance of these alloys by investigations of the dissolved matrix around typical by using the electrolytic extraction process.

## 2. Materials and Methods

In order to investigate the effect of inclusions on corrosion properties, two commercially available nickel-based alloys were selected. The materials were produced by vacuum induction melting and vacuum remelting processes and then forged and heat treated (solution annealing than stepped ageing) according to corresponding specifications that are described in detail in [24]. The chemical compositions of the studied materials are given in Table 1.

**Table 1.** Chemical compositions (wt %) of the studied nickel-based alloys.

| Alloy | C Max | Ni | Cr | Mo | Nb | Ti | W | Al | P Max | S Max |
|---|---|---|---|---|---|---|---|---|---|---|
| EP718 | 0.1 | 43–47 | 14–16 | 4.0–5.2 | 0.8–1.5 | 1.8–2.4 | 2.5–3.5 | 0.9–1.4 | 0.01 | 0.015 |
| Alloy 718 | 0.045 | 50–55 | 17–21 | 2.8–3.3 | 4.8–5.2 | 0.8–1.2 | - | 0.4–0.6 | 0.01 | 0.01 |

The method of electrolytic extraction (EE) was used for 3D investigations of inclusions. The extraction was done using a 10% AA electrolyte (10% acetylacetone–1% tetramethyl-ammonium chloride-methanol) and using the following parameters: 0.04–0.06 A an electric current and 2.9–3.8 V a voltage. The temperature of electrolyte during extraction was ambient.

According to previous studies [32], this electrolyte was selected because it does not dissolve the detected inclusions, but it dissolves the matrix. Therefore, it can be assumed that the inclusions were not directly involved in the electrochemical dissolution of the matrix around the inclusion. During a typical 150–180 min of extraction approximately 0.08–0.10 g of alloy samples were dissolved during each electrolytic extraction, which correspond to a dissolved metal layer of 70–95 μm. After dissolution,

a filtration of the solution was made where the inclusions were collected on a surface of the polycarbonate film filter, having a hole diameter of 0.4 µm. Each test was duplicated.

Characteristics of non-metallic inclusions (such as composition, morphology, size, number and location) were analyzed by means of scanning electron microscopy (SEM) (Hitachi, Tokio, Japan) equipped with energy dispersive spectroscopy (EDS). The inclusions were investigated on the surface of film filters (FF) as well on the surface of metal samples (MS) that had been subjected to dissolution. Some SEM images of typical inclusions investigated on a film filter and metal surface after EE are shown in Figure 1.

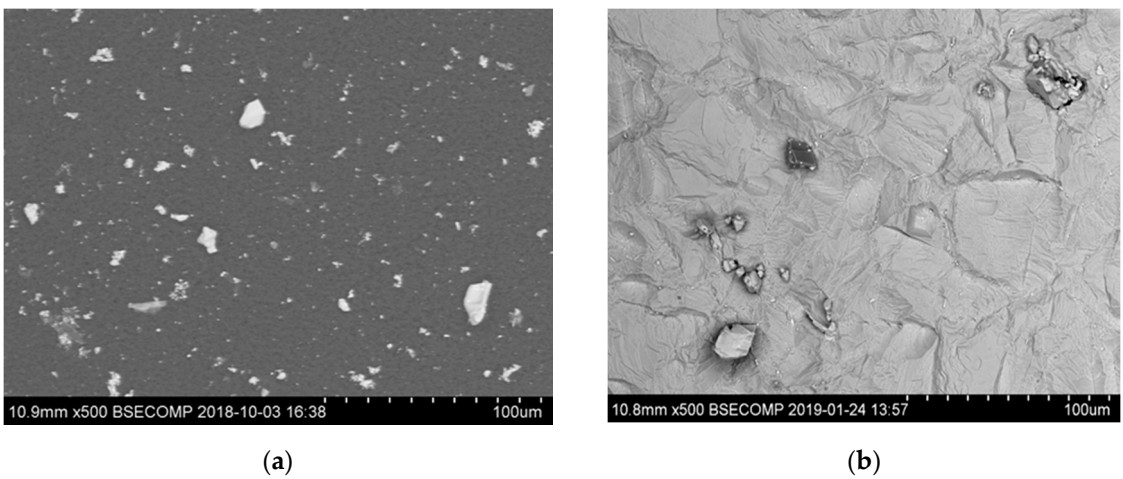

(**a**)  (**b**)

**Figure 1.** Typical non-metallic inclusions found on film filter (**a**) and metal surface (**b**) after EE.

For the evaluation of the effect of inclusions on the corrosion resistance of alloy, a dissolution extent of the matrix around different inclusions after electrolytic extraction was determined by measuring the diameters of the crater ($D_{cr}$) around inclusions and by calculating the relative coefficient of metal dissolution ($KD$), using the following relationships:

$$D_{cr} = \sqrt{(4 \times A_{cr}/\pi)} \tag{1}$$

$$KD = A_{cr}/A_{incl} \tag{2}$$

where $A_{cr}$ and $A_{incl}$ are the area of the crater around an inclusion and the area of this inclusion measured on a SEM image, respectively.

## 3. Results

### 3.1. Characterization of Inclusions after Electrolytic Extraction of Metal Samples

Based on the 3D investigations of inclusions on film filters and on surfaces of metal samples after electrolytic extractions, it was found that the total number of inclusions per unit volume in an EP718 is about 6.5 times larger than that in a 718 alloy. However, in most cases, a determination of only the total content of all inclusions cannot explain the difference in corrosion resistance of similar steels or alloys. It is well known that different inclusions (having various characteristics such as compositions, morphology, size and location in the matrix) could lower the corrosion properties of steels and alloys. Therefore, this study was focused on the estimation of the effect of these characteristics of different types of inclusions on the dissolution extent of the matrix around these inclusions during electrolytic extraction. This behavior is believed to be similar to the corrosion processes that occur in the most prominent industrial environments, including oil and gas wells.

The different inclusions observed in 3D on the film filter and on the metal surface after electrolytic extraction of the EP718 sample were divided into four groups, based on their composition and morphology.

Table 2 shows typical SEM images, contents of the main elements and size ranges of different types of inclusions. Since the contents of such light elements as C, N and O cannot be accurately determined by EDS, their contents have not been determined.

**Table 2.** Typical inclusions observed after electrolytic extraction of an EP718 sample.

| Type | SEM Image on Film Filter | SEM Image on Metal Surface | Composition (Mass. %) | Size (μm) |
|---|---|---|---|---|
| NbTi-C (inclusions and clusters) |  |  | 50–70% Nb, 24–45% Ti, 1–6% W | 3–40 |
| TiNb-N, TiNb-NC (inclusions and clusters) |  |  | 48–83% Ti, 2–42% Nb, 0–5% W | 4–26 |
| TiNb-S, TiNb-SC (inclusions and clusters) |  |  | 40–71% Ti, 14–33% Nb, 0–4% W, 0–3% Mo, 7–21% S | 2–37 |
| NbTiMoWCr-C |  |  | 8–57% Nb, 3–42% Ti, 0–38% Mo, 0–26% W, 3–25% Cr | 0.5–12 |

It was found that the sample of EP718 alloy contains the following typical inclusion types:

1.  Irregular or regular carbides containing Nb and Ti as well as up to 1–6% of W (NbTi-C type) in the size range of 3–40 μm. The ratio of Nb and Ti of these carbides ($R_{Nb/Ti}$ = %Nb/%Ti) in these inclusions varied from 1.2 up to 2.4 with an average value of 1.6 ± 0.2.
2.  Regular Ti and Nb nitrides (TiNb-N type) also contained up to 5% of W and had a size of 4–26 μm. Moreover, sometimes these inclusions contained small amounts of carbides of these elements as well. The values of a $R_{Nb/Ti}$ in these nitrides varied from 0.2 up to 0.6 (0.3 ± 0.2 on average) depending on the fraction of NbTi-C inclusions precipitated on nitrides.
3.  Irregular Ti and Nb sulphides (TiNb-S type) also contained 7–21% of S, ≤4% of W and ≤3% of Mo. These sulfides are 2–37 μm in size and they were observed mostly as clusters on grain boundaries of a metal sample along the deformation direction in combination with carbides. The values of a

$R_{Nb/Ti}$ in these sulphides varied from 0.2 up to 0.7 (0.5 ± 0.2 on average) depending on the fraction of carbide inclusions. Moreover, it was found that these types of sulphides, that were detected as acicular inclusions by using a conventional 2D investigations on polished surface of metal samples, have a plate or petal-like shape with thickness 1–2 μm.

4.  Small size multicomponent carbides (0.5–4 μm) containing Nb, Ti, Mo, W and Cr and having different morphologies (such as spherical, irregular and acicular shapes). Length of acicular inclusions can be up to 12 μm. The contents of the main elements in these inclusions may vary ($R_{Nb/Ti}$ = 1.4 ± 0.5). These types of inclusions were observed mostly on grain boundaries of the matrix and partially on surfaces of different inclusions. Based on the location, morphology and compositions, it can be safely assumed that these inclusions precipitated during solidification of a matrix during cooling and during heat treatment.

It should be noted that the NbTi-C and TiNb-N inclusions observed on the film filter and on the metal surface after electrolytic extraction were found to be separate particles as well as clusters or groups of particles located very close to each other, which summarized size is significantly larger than that of separate particles. Moreover, sometimes the inclusions of different types were observed together in one group or located close to each other.

Based on the composition and morphology, the inclusions observed in alloy 718 were divided into the three groups (as shown in Table 3).

**Table 3.** Typical inclusions observed after electrolytic extraction of a 718 alloy.

| Type | SEM Image on Film Filter | SEM Image on Metal Surface | Composition (Mass. %) | Size (μm) |
|---|---|---|---|---|
| NbTi-C (inclusions and clusters) | | | 72–96% Nb, 2–16% Ti, 0–7% Cr | 2–30 |
| TiNb-N | | | 59–79% Ti, 9–39% Nb | 9–27 |
| NbTiCr-C | | | 78–93% Nb, 6–15% Ti, 1–3% Cr | 0.5–8 |

It was found that the sample of alloy 718 contains the following typical inclusions:

1.  Regular and irregular carbides containing Nb and Ti (NbTi-C type) and also containing 0–7% of Cr and having 2–30 μm sizes. The ratio of the Nb and Ti contents of these carbides ($R_{Nb/Ti}$) in these inclusions varied in a wide range (from 3.4 up to 23) and had an average value of 9.9 ± 3.7. These NbTi-C inclusions were observed on film filters and metal surfaces as separate particles as well as clusters or groups of particles located very close to each other, as shown in Figure 2.

2. Large size irregular nitrides containing Ti and Nb (TiNb-N type) having sizes ranging from 9 to 27 μm in size. The value of the $R_{Nb/Ti}$ ratio in these nitrides varied from 0.2 up to 0.9 μm (0.3 ± 0.2 on average). These nitride inclusions were usually located in the matrix as separate particles.

3. Small size carbides (0.5–8 μm) containing mostly Nb, Ti and 1–3% Cr (NbTi-C) and having spherical, irregular or acicular shapes, as shown in Figure 2. The value of the $R_{Nb/Ti}$ ratio in these nitrides varied from 5.6 up to 14.9 μm (8.6 ± 2.1 μm on the average). These small size carbides were usually located on grain boundaries of the matrix and sometimes on surfaces of different inclusions. It was assumed that these small carbides precipitated during solidification of in a solidified the matrix as well as during heat treatment.

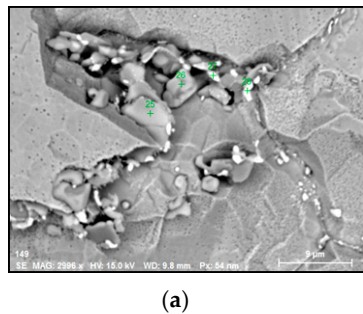 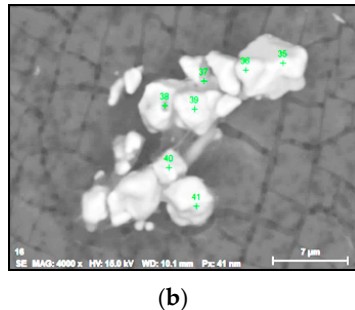 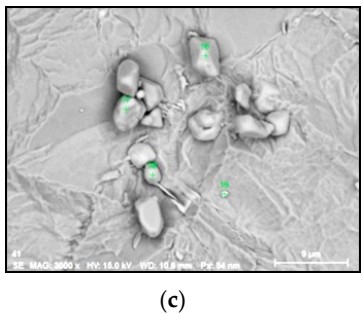

(**a**)　　　　　　　　　　　　　　　　(**b**)　　　　　　　　　　　　　　　　(**c**)

**Figure 2.** Typical SEM images of NbTi-C clusters on the metal surface in an EP718 alloy (**a**) and in a alloy 718 on a film filter (**b**) and of the group of NbTi-C inclusions located closely each other on metal surface (**c**) observed after EE.

### 3.2. Effect of Inclusions on Corrosion Resistance

After electrolytic extraction, the surfaces of metal samples were investigated for evaluation of the characteristics and location of different inclusions in the matrix. It was found that the small size inclusions (such as NbTiMoWCr-C in EP718 alloy and NbTiCr-C in alloy 718) and sulphides (TiNb-S and TiNb-SC in the EP718 alloy) are mostly located on grain boundaries of the matrix. However, large size nitrides and carbides in these Ni-based alloys can be observed both on grain boundaries as well as inside metal grains.

It was detected that the matrix was not evenly dissolved during the electrolytic extraction process around different inclusions. As can be seen in Figure 3, the dissolution of the matrix around some inclusions was almost uniform (Figure 3a) as in most areas of a metal sample, whereas large size "craters" of metal dissolution were observed around other inclusions and clusters (Figure 3b). Moreover, significant metal dissolutions present as "ditches" were observed on grain boundaries of the EP718 sample. These craters and ditches formed around some inclusions and on grain boundaries can be explained by the easier electro-chemical dissolution of the weakened matrix around these corrosion-active inclusions and on grain boundaries during electrolytic extraction. However, according to publications of Inoue et al. [32], the 10% AA electrolyte used in this study for electrolytic extraction does not dissolve the inclusions observed in these alloy samples. In other words, these inclusions are not directly involved in the dissolution process of the matrix around these inclusions due to some additional electro-chemical reactions. As a result, it can allow us to assess the corrosion resistance of the matrix around different inclusions and to detect the inclusions having the most harmful effects on weakening of the matrix around these inclusions.

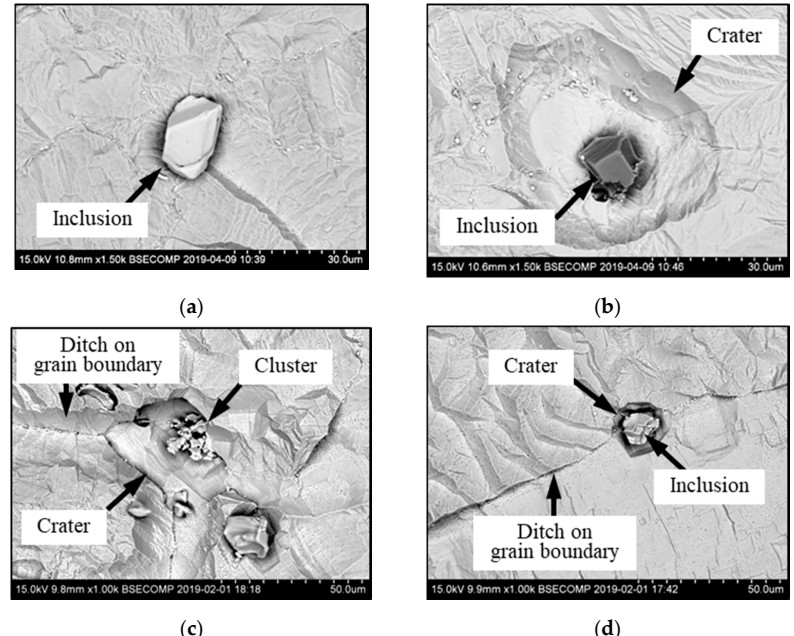

**Figure 3.** Typical SEM images of metal dissolution around different inclusions and clusters observed on the surface of a metal sample after electrolytic extraction. (**a**) a uniform dissolution around inclusion, (**b**) dissolution inclusions on grain boundaries, (**c**) dissolution around craters, (**d**) craters around inclusion.

In order to clarify the degree of influence of various inclusions on the corrosion resistance, the dissolution of the matrix near the inclusions on the surface of the metal samples was analyzed after electrolytic extraction. The extent of the matrix dissolution around various inclusions after electrolytic extraction was estimated and compared by determining the diameter of the dissolution crater around inclusions ($D_{cr}$, Equation (1)) as well as the relative coefficient of metal dissolution ($KD$, Equation (2)). A comparison of some characteristics of different inclusions, size ranges of craters ($D_{cr}$) and average $KD$ values (including value of standard deviation) in investigated alloy samples are given in Table 4.

**Table 4.** Characteristics of different inclusions and dissolution of the matrix around inclusions obtained for different alloy samples.

| Alloy | Type of Inclusion | Average $R_{Nb/Ti}$ (= %Nb/%Ti) | Size Range of Inclusions (μm) | Size of Crater, $D_{cr}$ (μm) | Average $KD$ |
|---|---|---|---|---|---|
| EP718 | NbTi-C | 1.6 ± 0.2 | 3–40 | 6–24 (16–40) * | 2.4 ± 1.1 (4.9 ± 3.7) * |
| | TiNb-N | 0.3 ± 0.2 | 4–26 | 9–10 (-) | 2.8 ± 0.9 (-) |
| | TiNb-S | 0.3 ± 0.2 | 2–37 (-) | - (15–40) | - (8.8 ± 3.6) |
| | NbTiMoWCr-C | 1.4 ± 0.5 | 0.5–12 | - (8–12) | - (21.3 ± 7.7) |
| Alloy 718 | NbTi-C | 9.9 ± 3.7 | 2–30 | 6–21 (11–40) | 2.2 ± 0.8 (5.1 ± 2.7) |
| | TiNb-N | 0.3 ± 0.2 | 9–27 | 17–26 (24–36) | 2.4 ± 0.1 (6.5 ± 5.5) |
| | NbTiCr-C | 8.6 ± 2.1 | 0.5–8 | - (2–7) | - (5.2 ± 3.2) |

*—Values for inclusions located at grain boundaries are indicated in parentheses.

Based on the obtained results, relationships between sizes of different inclusions, their locations in the matrix and dissolution parameters ($KD$ and $D_{cr}$) were shown determined as in Figures 4 and 5 for EP718 alloy and alloy 718, respectively.

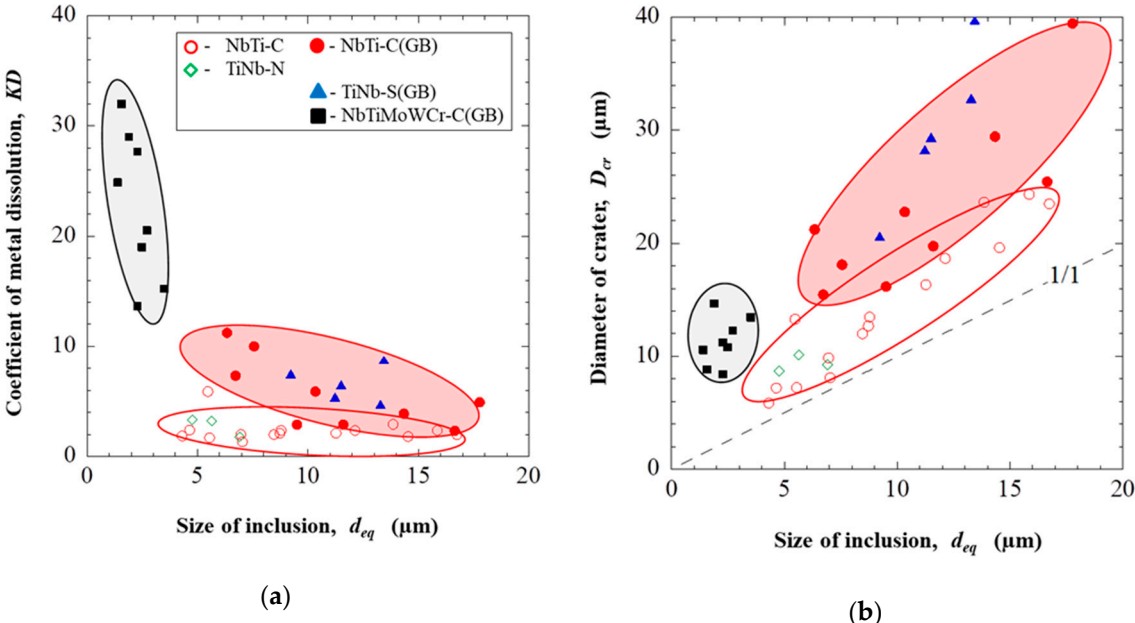

(**a**)　　　　　　　　　　　　　　　　　　　　　　　(**b**)

**Figure 4.** Effect of the composition, size and location of inclusions in the EP718 alloy on the dissolution coefficient (**a**) and on the diameter of crater around inclusions (**b**).

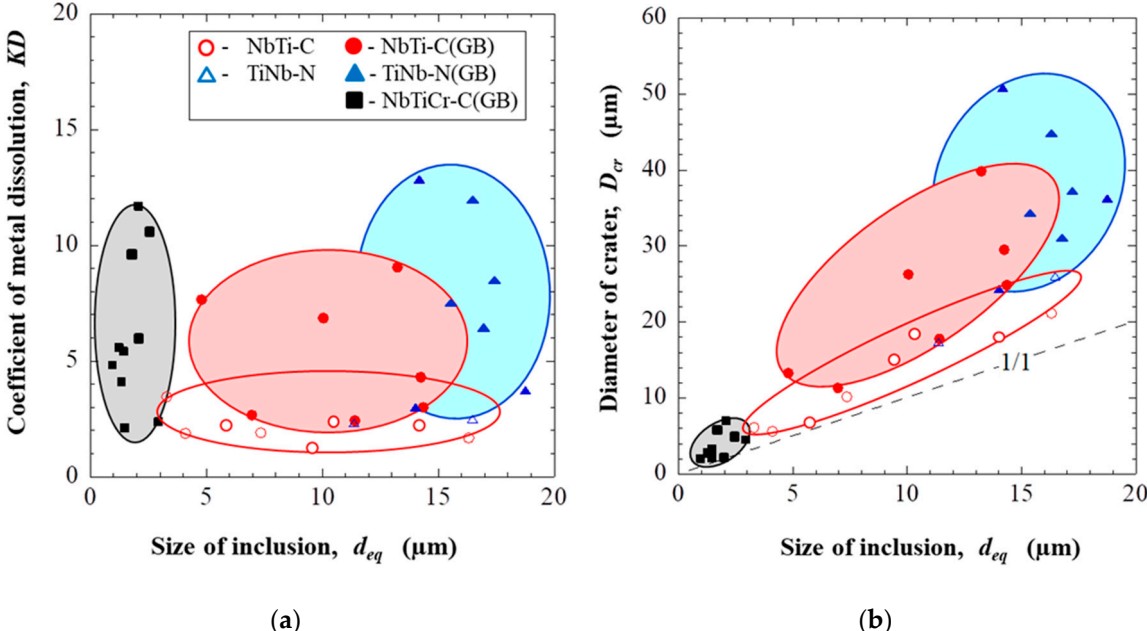

(**a**)　　　　　　　　　　　　　　　　　　　　　　　(**b**)

**Figure 5.** Effect of the composition, size and location of inclusions in the alloy 718 on the dissolution coefficient (**a**) and on the diameter of crater around inclusions (**b**).

According to the obtained results given in Table 4 and Figure 4, the largest sizes of craters ($D_{cr}$ up to 40 μm) in the EP718 sample were found close to NbTi-C and TiNb-S inclusions of titanium carbides, niobium, and titanium sulphides located along grain boundaries. The average dissolution coefficients of the matrix around these inclusions ($KD$) are 4.9 and 8.8, respectively, this suggests that sulphides are more active in terms of corrosion compared to other inclusion types. Titanium sulphides, which are

rarely mentioned in the literature, were also detected in the EP718 alloy. Nevertheless, it is known that sulphides are corrosive inclusions that significantly reduce the pitting resistance (from 3 to 100 times) and that they are preferred centers for pitting, especially in chloride environments [5,9].

It should be pointed out that the NbTi-C inclusions located inside of grains have 2 times smaller values of $D_{cr}$ and $KD$ values compared to inclusions located on grain boundaries. Moreover, it can be seen that the small amount of NbTiMoWCr-C inclusions located at the grain boundaries have the highest values of the dissolution coefficient $KD$ (from 14 up to 32) compared to the other types of inclusions. This can be explained by the depletion of the matrix around these inclusions by the alloying elements (such as chromium, molybdenum and tungsten), which are contained in these inclusions and responsible for corrosion resistance of the matrix. However, the sizes of the craters/ditches around these inclusions are relatively small (less than 15 μm) compared to the other types of inclusions, due to the smaller size of these inclusions.

The TiNb-N inclusions were observed inside the grains and, judging by the low dissolution coefficient ($KD \sim 2.8$) and small diameter of craters (9–10 μm), they have a lower effect on corrosion. These results are in agreement with published literature data [21].

As can be seen in Table 4 and Figure 5, the largest craters in the alloy 718 sample were observed around TiNb-N ($D_{cr}$ = 24–51 μm) and NbTi-C ($D_{cr}$ = 16–40 μm) inclusions located at the grain boundaries. The size of craters around these types of inclusions tends to increase significantly with an increased equivalent size ($D_{eq}$) of inclusions. Though the $KD$ coefficient for these inclusions vary in a wide range (from 2 up to 13), the dissolution coefficient of the matrix around these inclusions located on grain boundaries is on average 2.5 times larger compared to inclusions located inside of grains.

The values of $KD$ for small size NbTiCr-C inclusions, which are located mostly on grain boundaries, vary in a wide range (from 2 up to 12) although the size of craters/ditches around these inclusions in this 718 alloy usually is much smaller ($D_{cr}$ = 2–7 μm) compared to that of other types of inclusions. It is interesting to point out that these NbTiCr-C inclusions, which also contained chromium as a corrosion resistance alloying element, did not practically show any particular effects on the corrosion dissolution of the matrix in the 718 alloy. This may be explained by the much lower content of Cr (1–3%) in these small carbides in the 718 alloy compared to the contents of alloying elements (3–25% Cr, 0–38% Mo and 0–26% W) in NbTiMoWCr-C inclusions in the EP718 alloy. Therefore, the matrix around these small NbTiCr-C inclusions cause no (or a small) concentration depletion by the alloying elements (such as Cr). As a result, the dissolution of the matrix ($KD$) around these small inclusions on the grain boundaries in the 718 alloy is much smaller (~4 times) compared to that in the EP718 alloy.

## 4. Discussion

Based on a comparative analysis of inclusions and dissolution of the matrix around these inclusions after electrolytic extraction for alloys EP718 and 718 alloys, the effect of this inclusions to corrosion properties was determined.

Similar types of inclusions (such as NbTi-C and TiNb-N) which correspond primary to MC carbides based on the chemistry and morphology, and which are formed when solidifying [33,34] were found in both Ni-based alloys. For both studied nickel-based alloys, the dissolution of the matrix around of NbTi-C inclusions and clusters has very similar dissolution parameters as those for inclusions located inside the grains as well as for inclusions located on grain boundaries, as can be seen in Table 4. However, it should be pointed out that the carbides in the 718 alloy contain a much larger amount of Nb ($R_{Nb/Ti}$ = 9.9 ± 3.7) compared to the EP718 alloy ($R_{Nb/Ti}$ = 1.6 ± 0.2), which can be explained by the larger content of Nb in the alloy 718 (up to 5.2%).

It was found that TiNb-N inclusions that are MN nitrides in both samples have a very similar ratio of Nb and Ti ($R_{Nb/Ti}$ = 0.3 ± 0.2) [33,34]. However, despite the fact that the average dissolution coefficients ($KD$) for nitrides (TiNb-N) in both alloys are similar (2.8 in the EP718 alloy and 2.4 in the alloy 718), the equivalent diameter of "craters" in the alloy 718 is significantly larger ($D_{cr}$ = 17–26 μm) compared to that in the EP718 alloy ($D_{cr}$ = 9–10 μm). It can be explained by a larger amount of nitrides

observed in the alloy 718 samples, though some large sized nitride clusters also were found in the EP718 sample. Moreover, most of the large size TiNb-N inclusions in the 718 alloy are located on grain boundaries of the matrix, for which the dissolution parameters are much higher ($KD = 6.5 \pm 5.5$ and $D_{cr} = 24$–36 μm). Thus, despite the fact that nitrides are considered to be the most neutral from the point of view of corrosion, their large sizes (10–27 μm) can harmfully affect the corrosion resistance by increasing the area of the cathode.

Both alloys are also characterized by a more active harmful effect of all types of inclusions located along grain boundaries. The values of the dissolution coefficient $KD$ for inclusions located at grain boundaries are from 2.1 to 2.7 times larger than that for inclusions located in the grain for both investigated alloys. The dissolution coefficient of the matrix around TiNb-S inclusions and clusters ($KD = 8.8 \pm 3.6$), which were found on grain boundaries in the EP718 alloy, is considerably larger compared to that for NbTi-C and TiNb-N inclusions in this alloy. Therefore, this type of sulphides should be considered as very harmful inclusions, which can significantly increase the pitting corrosion of EP718 alloys.

The dissolution coefficient for small chromium-containing carbide inclusions located on grain boundaries is much lower for the alloy 718 ($KD = 5.2 \pm 3.2$) compared to that for the EP718 alloy ($KD = 21.3 \pm 7.7$). In contrast to the EP718 alloy, the $KD$ value for these small Cr-containing inclusions is relatively low and comparable to the dissolution coefficients for other inclusions observed in the alloy 718. This difference in behavior can be explained by the lower contents of Cr (1–3%) small carbides in 718 alloy and by the more complex composition of these carbide inclusions of the EP718 alloy (Table 2) specifically, they contain 3–25% of Cr, 0–38% of Mo and 0–26% of W, which corresponds to $M_6C$ carbides according to chemical composition and they are formed in the temperature range from 850 to 1100 °C [33,34]. These carbides are believed to be responsible for pitting resistance.

Based on the revealed differences in the compositions and amounts of inclusions in the EP718 and 718 alloys, it can be concluded what inclusion types causes a reduction of the corrosion resistance of EP718 alloy.

1. Titanium sulphides have the most negative effect on the matrix dissolution;
2. Carbides, especially at the boundaries, containing chromium, molybdenum and tungsten in their composition cause a significant dissolution of the matrix around the inclusions;
3. Large (more than 10 μm) carbides and nitrides of titanium and niobium are also able to reduce the corrosion resistance.

An application of the electrolytic extraction in combination with 3D investigations of different inclusions and clusters and an evaluation of the electrolytic dissolution of the matrix around these inclusions in the EP718 and 718 alloys allowed a detection of the most harmful inclusions with respect to the corrosion of the matrix in these Ni-based alloys. As a next step of the future study, the reasons of formations and mechanisms of effects of harmful inclusions on different stages of the Ni-based alloy production need to be considered.

## 5. Conclusions

In this study, the characteristics of different inclusions (such as composition, morphology, size, number and location) and dissolution of the matrix around these inclusions were investigated on film filters and on the surfaces of metal samples after electrolytic extraction (EE) of two industrial Ni-based alloys (alloy 718 and EP718). The obtained results can be summarized as follows:

1. The electrolytic extraction technique can successfully be applied for three-dimensional (3D) investigations of different inclusions on film filters and surfaces of metal samples after extraction of Ni-based alloys. It was shown that the morphology of inclusions is much more complicated than what can be determined on a flat section. For instance, a thin plate-like sulphides of Ti and Nb (with a thickness of 1–2 μm) that were detected in clusters (up to 37 μm) are located mostly on

grain boundaries in the EP718 alloy. However, these which were detected only as separate acicular sulphides by using conventional two-dimensional (2D) investigations on polished surfaces of this metal sample.

2.  An evaluation of different extents of dissolution of the matrix around different inclusions after EE by determination of equivalent diameter of "crater" ($D_{cr}$) and relative coefficient of the matrix dissolution (*KD*) makes it possible to estimate a metal weakening around investigated inclusions, which correlates to a corrosion resistance of metal.

3.  In the EP718 alloy, four types of inclusions were found. These are listed according to the higher degree of harmful influence on the corrosion resistance of the alloy: TiNb-S sulphides, NbTi-C carbides, small size multicomponent carbides (NbTiMoWCr-C) and TiNb-N nitrides. Three types of typical inclusions were found in alloy 718, namely NbTi-C carbides, TiNb-N nitrides and small size carbides (NbTiCr-C). In addition to separate inclusions, clusters (up to 40 µm) consisting of different inclusions were found in both alloys.

4.  The most harmful effects of inclusions on dissolution of the matrix were found to occur for the sulphides (TiNb-S) and small carbides (NbTiMoWCr-C) located on grain boundaries. The large carbides (NbTi-C) and nitrides (TiNb-N) located as on grain boundaries as well inside of grains have less harmful influence. The nitride inclusions (TiNb-N) having sizes larger than 10 µm can also significantly reduce the corrosion resistance of Ni-based alloys, although in the literature they are described as the most neutral with respect to the influence on the matrix.

5.  In addition to the composition of the inclusion, their location (at the boundary or in the grain) and size also affect the corrosion resistance and a pitting propagation. Inclusions located at the grain boundaries reduce the corrosion resistance the most. For instance, the dissolution parameters (*KD* and $D_{cr}$) for inclusions located on grain boundaries are from 2.1 to 2.7 times larger than those for inclusions located inside the grains for both investigated alloys. Large inclusions of more than 10 µm affect the corrosion resistance more significantly even if they mostly are neutral nitrides inclusions.

**Author Contributions:** Conceptualization, methodology, investigation, draft writing—A.K., E.A.; resources, validation, supervision, review and editing—P.G.J., A.A.; All authors have read and agreed to the published version of the manuscript.

**Funding:** This research received no external funding.

**Conflicts of Interest:** The authors declare no conflict of interest.

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
