# Peer review of "Effect of Inclusions on the Corrosion Properties of the Nickel-Based Alloys 718 and EP718"

_metals, doi:10.3390/met10091177_

Round 1

Reviewer 1 Report

This work describes the inclusionary health of two nickel – based alloys using electrochemical dissolution, filtering and SEM observation and shape/dimensional characterization.  The presented examples are interesting and of importance in industry.

The manuscript is rather clear and globally well written, despite the presence of too many grammar mistakes and link problems between text and figures/tables. It should be re-written/re-prepared more carefully.

Additionally I suggest the following change everywhere in the text: please avoid using “metal matrix”; only “matrix” or “alloy’s matrix” is enough and does not evocate composites materials

Some examples below:

Page 01/Lines 10-11 “lower deformation, 10 mechanical, corrosion and other properties.” -> “lower resistance to deformation, mechanical behavior, corrosion resistance and other properties.”

P01/L11 “on film filter” what does this mean? P02/L81: OK, understand. Nevertheless text must be rewritten in this location of the summary

P01/L16-19 and elsewhere: “TiNb-C” or “TiNb-N” may be advantageously replaced by “(Ti,Nb)C” and “(Ti,Nb)N”

P01/L32 “and alloys” -> “and other alloys”; problem with two references; there are other reference problems farer in the text

P01/L40 “widely spread” -> “are widely spread”

P02/L70 “according” -> “according to”

P02/L71 Sentence not finished; followed by “Table 1.” I presume?

P02/L73 “The method of…”

P02/L75 “2.9-3.8 V According” -> “2.9-3.8 V. According”

P03/L88 “For the evaluation…”

P03/L93 “on an SEM” -> “on a SEM”

P03/L101 please re-write “can significantly effect on the corrosion”

P03/L102 “on estimation of the effect” -> “on the estimation of the effect” / I stop here correcting the absence of “the” in your sentences; please apply these corrections over the full text

P03/L110 please correct “their contents did not give in the Table 2.”

P04 some informations in Table 2 and in the following text are redundant; please rearrange

P08/L196-198 = P08/L199-201; please be more careful for your manuscript before submitting it…

Caution to grammar errors too: close to the end of the manuscript (P09/L250-251: “which correspond primary MC carbides” etc...)

Author Response

Dear reviewer, thank you for your work and for your attention to our article. Please consider replies to your comments.

Please note that some of the lines could have changed relatively to the ones you specified.

“metal matrix” was corrected to only “matrix” throughout the text.

Point 1: Page 01/Lines 10-11 “lower deformation, 10 mechanical, corrosion and other properties.” -> “lower resistance to deformation, mechanical behavior, corrosion resistance and other properties.”

Response 1: corrected

Point 2: P01/L11 “on film filter” what does this mean? P02/L81: OK, understand. Nevertheless text must be rewritten in this location of the summary

Response 2: the phrase has been clarified “determined for inclusions collected on film filters after electrolytic extraction”

Point 3: P01/L16-19 and elsewhere: “TiNb-C” or “TiNb-N” may be advantageously replaced by “(Ti,Nb)C” and “(Ti,Nb)N”

Response 3: we use more general designation “TiNb-C” or “TiNb-N” to avoid mistakes since we did not clarify the exact ratio of the elements in this inclusions. And if for “(Ti,Nb)C” and “(Ti,Nb)N” it could be more confident using designation MC but for M23C6, M6C could be mistake.

Point 4: P01/L32 “and alloys” -> “and other alloys”; problem with two references; there are other reference problems farer in the text

Response 4: corrected.

References were corrected as well as other in the text.

Point 5: P01/L40 “widely spread” -> “are widely spread”

Response 5: corrected

Point 6: P02/L70 “according” -> “according to”

Response 6: corrected

Point 7: P02/L71 Sentence not finished; followed by “Table 1.” I presume?

Response 7: “The chemical composition of the studied materials is given in Table 1.”

Point 8: P02/L73 “The method of…”

Response 8: corrected

Point 9: P02/L75 “2.9-3.8 V According” -> “2.9-3.8 V. According”

Response 8: “.” added

Point 10: P03/L88 “For the evaluation…”

Response 10: corrected

Point 11: P03/L93 “on an SEM” -> “on a SEM”

Response 11:corrected

Point 12: P03/L101 please re-write “can significantly effect on the corrosion”

Response 12: corrected written “could lower corrosion properties of steels and alloys”

Point 13: P03/L102 “on estimation of the effect” -> “on the estimation of the effect” / I stop here correcting the absence of “the” in your sentences; please apply these corrections over the full text

Response 13: corrected. The whole text was checked by our specialists.

Point 14: P03/L110 please correct “their contents did not give in the Table 2.”

Response 14: their contents has not been determined.

Point 15: P04 some information in Table 2 and in the following text are redundant; please rearrange

Response 15: Please clarify what information is redundant

Point 16: P08/L196-198 = P08/L199-201; please be more careful for your manuscript before submitting it…

Caution to grammar errors too: close to the end of the manuscript (P09/L250-251: “which correspond primary MC carbides” etc...)

Response 16: repeat removed

 “.” Was added

Reviewer 2 Report

NO comments.

Author Response

reviewer 2 pointed out another publication, see more at: https://www.e3s-conferences.org/articles/e3sconf/abs/2019/47/e3sconf_cr18_04004/e3sconf_cr18_04004.html Characterization of non-metallic inclusions in corrosion -resistance nickel -
based EP718 and 718 alloys by using electrolytic extraction method. Andrey
Karasev1*, Ekaterina Alekseeva2, Aleksey Lukianov3 and Pär G. Jönsson1.
Could you please explain the novelty of this new paper?

Answer: The article link to our conference paper “Characterization of non-metallic inclusions in corrosion-resistance nickel-based EP718 and 718 alloys by using electrolytic extraction method” was added on the first page. The present  article (“Effect of inclusions on the corrosion properties of nickel based alloys 718 and EP718") is an extended version of before mentioned conference paper. It contains more detailed  and systematical information: the introduction with the literature review, the procedure for assessing the effect of inclusions on corrosion is described in detail, the results including new pictures of separate inclusions and clusters for in the EP718 and Inconel 718 samples. Also, the results and obtained tendencies were analyzed again based on repeated and more systematical performed experiments investigations. The present article contains more precise co characteristics results of the different inclusions and dissolution of matrix around inclusions obtained for both alloys. Additionally, dependencies of the effect of the composition, size and location of inclusions on the dissolution coefficient and diameter of crater around inclusions for both alloys.

Reviewer 3 Report

The paper is written very well and authors clearly state what was achieved in their studies. Despite this, there are some issues that need to be corrected.

Page 1, line 32 - missing reference

Page 2, table 1 - missing alignment to text, table to wide, line 73 missing empty line below the table

Page 5 Micrographs in table 3 alignment

Page 6 line 162 red letter "a"

Page 8 and 9 Error! Reference source not found x 3

Author Response

Dear reviewer, thank you for your work and for your attention to our article. Please consider replies to your comments.

Point 1: Page 1, line 32 - missing reference

Response 1: corrected

Point 2: Page 2, table 1 - missing alignment to text, table to wide,  line 73 missing empty line below the table

Response 2: aligned, corrected

Point 3: Page 5 Micrographs in table 3 alignment

Response 3: Micrographs were divided.

Point 4: Page 6 line 162 red letter "a"

Response 4: corrected

Point 5: Page 8 and 9 Error! Reference source not found x 3

Response 5 : corrected

Reviewer 4 Report

First sentence in the introduction is too long, consider breaking up.

Line 32. The sentence starting with "Main" need rewording. Do you mean "the voids and stressed zones between the intermetallics and the matrix are the sites for microcrack initiation and propagation 

Experimental - please specify the heat treatment the samples underwent 

What was the temperature of the electrolyte and the time for extraction 

Were repeats performed of the EE on each sample, if not, they should be so the variability in intermetallic distribution can be understood.

line 196 -201 - there is a repeated paragraph

Is this classic sensitisation where the carbides tie up protective elements such as chromium and therefore lead to a depletion in the surrounding matrix and a corresponding reduction in corrosion resistance?

This work would benefit from some SKP-AFM experiments to show relative activity of the intermetallics. 

Author Response

Response to Reviewer 4 Comments

Dear reviewer, thank you for your work and for your attention to our article. Please consider replies to your comments.

Point 1: First sentence in the introduction is too long, consider breaking up.

Response 1 : The first sentence was divided into two shorter sentences.

Point 2: Line 32. The sentence starting with "Main" need rewording. Do you mean "the voids and stressed zones between the intermetallics and the matrix are the sites for microcrack initiation and propagation 

Response 2 : line 33-34 Changed to «Voids and stressed zones between the inclusion and the matrix are the sites for microcrack and pitting initiation»

Point 3: Experimental - please specify the heat treatment the samples underwent 

Response 3 : Heat treatment added on the page 2/ line89-90 : solution annealing and ageing.

Point 4: What was the temperature of the electrolyte and the time for extraction 

Response 4: Temperature of electrolyte was around ambient. Added in line 95. Time of extraction added in line 99.

Point 5: Were repeats performed of the EE on each sample, if not, they should be so the variability in intermetallic distribution can be understood.

Response 5: added in line 103 “Each test was duplicated”

Point 6: line 196 -201 - there is a repeated paragraph

Response 6 : repeated paragraph was eliminated

Point 7: Is this classic sensitization where the carbides tie up protective elements such as chromium and therefore lead to a depletion in the surrounding matrix and a corresponding reduction in corrosion resistance?

Response 7 : In this work there is an effort to show that not only Cr,Mo,W carbides (“classic sensitization”) tend to lower corrosion resistance due to depletion but also MC, MN, etc. particles are able to lower corrosion resistance due to other mechanism that to be clarified.

Point 8: This work would benefit from some SKP-AFM experiments to show relative activity of the intermetallics. 

Response 8 : Thank you for your advice!

Round 2

Reviewer 4 Report

Interesting work and I am happy with the changes made to the paper.